# Simple predictive models identify patients with COVID-19 pneumonia and poor prognosis

Mar Riveiro-Barciela[1,2,3], Moisés Labrador-Horrillo🄌[3,4,5]*, Laura Camps-Relats[1], Didac González-Sans[6], Meritxell Ventura-Cots[1,2], María Terrones-Peinador[6], Andrea Nuñez-Conde[6], Mónica Martínez-Gallo[3,7,8], Manuel Hernández[3,7,8], Andrés Antón[3,9], Antonio González[6], Ricardo Pujol-Borrell[3,7,8], Fernando Martínez-Valle[3,6]

1 Liver Unit, Internal Medicine Department, Hospital Universitario Vall d'Hebrón, Vall d'Hebron Barcelona Hospital Campus, Barcelona, Spain, 2 Centro de Investigación Biomédica en Red de Enfermedades Hepáticas y Digestivas (CIBERehd), Instituto de Salud Carlos III, Madrid, Spain, 3 Universitat Autònoma de Barcelona, Barcelona, Spain, 4 Allergy Section, Internal Medicine Department, Hospital Universitario Vall d'Hebrón, Vall d'Hebron Barcelona Hospital Campus, Barcelona, Spain, 5 ARADyAL Research Network Instituto de Salud Carlos III, Madrid, Spain, 6 Systemic Autoimmune Diseases Unit, Internal Medicine Department, Hospital Universitario Vall d'Hebron, Vall d'Hebron Barcelona Hospital Campus, Barcelona, Spain, 7 Diagnostic Immunology group, Vall d'Hebron Institut de Recerca (VHIR), Barcelona, Spain, 8 Division of Immunology, Department of Cell Biology Physiology and Immunology, Hospital Universitari Vall d'Hebron, Vall d'Hebron Barcelona Hospital Campus, Barcelona, Spain, 9 Respiratory Viruses Unit, Microbiology Department, Hospital Universitari, Vall d'Hebron Vall d'Hebron Institut de recerca (VHIR), Barcelona Hospital Campus, Barcelona, Spain

* mlabrador@vhebron.net

**Data Availability Statement:** All relevant data are within the manuscript and its Supporting Information files.

## Abstract

### Background and aims

Identification of SARS-CoV-2-infected patients at high-risk of poor prognosis is crucial. We aimed to establish predictive models for COVID-19 pneumonia severity in hospitalized patients.

### Methods

Retrospective study of 430 patients admitted in Vall d'Hebron Hospital (Barcelona) between 03-12-2020 and 04-28-2020 due to COVID-19 pneumonia. Two models to identify the patients who required *high-flow-oxygen-support* were generated, one using baseline data and another with also follow-up analytical results. Calibration was performed by a 1000-bootstrap replication model.

### Results

249 were male, mean age 57.9 years. Overall, 135 (31.4%) required *high-flow-oxygen-support*. The baseline predictive model showed a ROC of 0.800 based on: SpO2/FiO2 (adjusted Hazard Ratio-aHR = 8), chest x-ray (aHR = 4), prior immunosuppressive therapy (aHR = 4), obesity (aHR = 2), IL-6 (aHR = 2), platelets (aHR = 0.5). The cut-off of 11 presented a specificity of 94.8%. The second model included changes on the analytical parameters: ferritin

**Funding:** This work was supported in part by grant PI20/00416 from the Instituto de Salud Carlos III and was co-financed by the European Regional Development Fund (ERDF). Principal investigator: Ricardo Pujol-Borrell The funders had no role in study design, data collection and analysis, decision to publish, or preparation of the manuscript.

**Competing interests:** No authors have competing interests.

(aHR = 7.5 if ≥200ng/mL) and IL-6 (aHR = 18 if ≥64pg/mL) plus chest x-ray (aHR = 2) showing a ROC of 0.877. The cut-off of 12 exhibited a negative predictive value of 92%.

## Conclusions

SpO2/FiO2 and chest x-ray on admission or changes on inflammatory parameters as IL-6 and ferritin allow us early identification of COVID-19 patients at risk of *high-flow-oxygen-support* that may benefit from a more intensive disease management.

## Introduction

COVID-19, the disease caused by the novel virus named severe acute respiratory syndrome coronavirus 2 (SARS-CoV-2) has emerged as a global public health threat. Since December 2019 [1], more than 31 million cases have been reported worldwide. The majority of cases are asymptomatic or with mild symptoms. However, a small proportion of infected subjects develop a severe pulmonary disease that can lead to acute respiratory distress syndrome, acute respiratory failure, multiple organ dysfunction syndrome and finally death [2]. In Europe, several countries such as UK, France, Italy and Spain have been severely affected, with epidemic outbreaks since the second half of February 2020. By September 3rd, 2020, 29.234 deaths had been reported in Spain [3].

Multiorgan damage might appear in severe COVID-19 cases. This syndrome is triggered by a dysfunctional host immune response to the virus that may persist in a context similar to the cytokine release syndrome [4–6]. Increased plasma levels of different interleukins and chemokines have been reported in patients with COVID-19 but it has been difficult to identify a pattern associated with poor prognosis [7]. Among cytokines, elevated levels of IL-6 have been consistently reported in several COVID-19 patient cohorts, correlates with disease activity [8], and with lymphopenia [9] and low NK cells counts [10]. Currently, treatment of COVID-19 infection consists of supportive care, including invasive and non-invasive oxygen support. In addition, many patients have received off-label or compassionate-use therapies, like antiretrovirals, antiparasitic agents, anti-inflammatory drugs, and convalescent plasma. Different immunosuppressive therapies aimed at limiting immune-mediated damage and hyperinflammation in COVID-19 are at various phases of development [6, 8]. Despite the different treatment approaches, it is of paramount importance that such anti-inflammatory treatments are used in the early phases of the disease in order to stop the inflammatory cascade. Although many factors have been associated with a poor prognosis in patients with COVID-19 infection, including genetic and laboratory findings [8, 11], simple models based on baseline features could be extremely useful for early identification of patients at risk of developing severe complications.

This study aimed to establish predictive models for disease severity based on baseline clinical and laboratory parameters, and thus, early identification of patients with SARS-CoV2 pneumonia who may benefit from a more intensive management, in order to ameliorate the course of the disease and reduce mortality.

## Materials and methods

### Study population

We included all consecutive adult patients hospitalized at the Internal Medicine wards due to COVID-19 pneumonia at Vall d'Hebron University hospital in Barcelona between 03/12/2020

and 04/28/2020. All patients had confirmed SARS-CoV-2 infection by RNA polymerase chain reaction (rtPCR) of nasopharyngeal swabs. In all cases demographic, clinical history and symptoms, radiological and laboratory data were collected from electronic medical records. All tests were performed according to the clinical care needs of the patient. Obesity was defined according to the WHO criteria as a BMI higher than 30. Radiologic assessment included always chest radiography and, in some cases, thoracic computed tomography (CT). Radiologic abnormality was determined by reviewing radiologic images by two attending physicians who extracted the data. In case of disagreement between the two reviewers a third physician reviewed the images. Radiologic findings were classified into: peripheral infiltrate, unilateral pneumonia and bilateral pneumonia. Unilateral pneumonia was defined if an infiltrate was observed in only one of the lungs. Peripheral infiltrate was described when involvement of the subpleural area was mainly present. Bilateral pneumonia was considered if diffuse infiltrates were present in both hemithorax.

Due to the complex situation with limited resources during the pandemic peak, number of patients admitted to the intensive care unit or with access to non-invasive mechanical ventilation or nasal high-flow cannula (NHFC) was restricted. For this reason, in order to ascertain the prognosis of patients, and based on the view that some treatments such as corticosteroids are only beneficial for patients who need respiratory support [12], we generate a composite variable based on the necessity of *high-flow-oxygen support*. This variable included all patients who required one or more of these conditions in order to maintain oxygen saturation over 90%: invasive mechanical ventilation, non-invasive mechanical ventilation, NHFC and non-rebreathing oxygen mask with reservoir. These mechanisms were used in a sequential way to obtain oxygen saturation over 90% in the case that the patient required a FiO2 higher than 50%.

## Laboratory assessments

Laboratory findings included complete blood count, electrolytes, liver and renal function tests and acute phase reactants including erythrocyte sedimentation rate (ESR), C-reactive protein (CRP), fibrinogen, D-dimer, lactate dehydrogenase (LDH), ferritin and IL-6. Blood tests were performed at admission, and at days 3 to 5, and 7 to 10 after admission in case of prolonged hospitalization. IL-6 was determined to indicate the use of some treatments, mainly Tocilizumab (TCZ).

Treatment of severe COVID19 has changed in our center during the course of the pandemic and adapted to the most up-to-date clinical evidence. At the beginning of the pandemic the majority of patients received lopinavir/ritonavir (LPV/r) and hydroxychloroquine (HCQ) alone or in combination. In some patients with severe disease, immune modulatory drugs were used empirically, mainly corticosteroids and/or anti-IL-6R (TCZ). Pulses of methylprednisolone were defined as doses ≥1 mgr/kg. TCZ was considered if IL-6 was higher than 40pg/dL and SpO2/FiO2 (SaFi) below 250 or in the event of clinical worsening. A single intravenous dose of 400 or 600mg of TCZ depending on the weight of the patient (≥75Kg) was administered. This study was approved by the Ethics Committee of the Vall d'Hebron Hospital (PR (AG)242/2020). Written informed consent was waived.

## Statistical analysis

Normally distributed quantitative variables were expressed as mean and standard deviation (SD), and compared with Student t test. Quantitative variables non-normally distributed were expressed as median and interquartile range (IQR) and analysed with the Mann-Whitney U test. Categorical variables were expressed as frequency and percentage and compared using

the chi-squared or Fisher's exact test, when frequencies were less than 5%. In order to help early identification of patients at high-risk of poor respiratory prognosis, assessment by the combined variable *high-flow-oxygen-support*, multivariate COX proportional hazard models was applied. We developed a model exclusively with baseline features. The second model included also follow up analytical data from day 3–5 after admission instead of baseline, for those patients who did not meet the outcome prior to that time. Variables showing a clinically and statistically significant association to the outcome in univariate analysis (Mantel-Cox-test) were selected for the initial models (p<0.10). The final models were fitted by using a step-wise forward method based on model Likelihood Ratios (Cox regression) with the same significance level (p<0.005) for entering and dropping variables. Quantitative variables included in the models were categorized (median or mean of the overall cohort, as appropriate) in order to increase the potency of the models. So as to estimate the performance of these models, calibration with a 1000-bootstrap samples was carried out to decrease the overfit bias [13, 14]. Finally, the model was converted into a weighted semiquantitative score. The discrimination performance of the new predictive models was evaluated by receiver operating characteristic (ROC) curve analysis and the concordance index (C-index). The cut-off values to differentiate patients with or without *high-flow oxygen support* was selected considering the highest Youden's index, and shown by the sensitivity, specificity and predictive values. The results were considered statistically significant when the p-value was lower than 0.05. All statistical analyses were performed using IBM SPSS, version 26.0 (SPSS Inc, Armonk, NY, USA).

### Ethics statement

Vall d'Hebron ethics committee approved the study for PI20/00416 project.

Oral consent was obtained from the patients in the context of the pandemic and recorded in their electronic clinical history. Patients' medical records were accessed between May to August of 2020 to obtain clinical data collection that was fully anonymized before analysed.

## Results

### Baseline characteristics of patients

Four hundred and thirty consecutive patients with pneumonia and COVID-19 infection confirmed by rtPCR were included. The majority of them were male (57.9%), mean age 57.9 years and 253 (58.8%) presented any comorbidities with arterial hypertension (39.3%) the most common. Obesity was present in 26.9% of the cohort, and underlying chronic obstructive pulmonary disease in 47 subjects (10.9%). At admission, 24 (5.6%) patients were on immunosuppressive therapy, mainly due to solid-organ transplant (Table 1).

The most frequent symptoms were fever in 342 (79.5%) and cough in 77.9% of patients. Median days from fever and cough onset to the admission were 6.3 and 7.1 days, respectively. Up to 57.9% of patients also complained of dyspnoea at admission. Other symptoms, like diarrhoea, myalgia and anosmia were less frequent (31.6%, 28.7% and 8.4%, respectively). All patients presented an abnormal chest x-ray, with peripheral infiltrate as the most common finding (46.1%), followed by bilateral pneumonia (36%) and unilateral pneumonia (17.9%).

At admission, 297 patients (69.1%) had an oxygen saturation over 95% and did not need oxygen support, whereas 54 patients (12.6%) required oxygen support over 0.31 litters per minute. Median ratio between oxygen saturation and oxygen requirement (SpO2/FiO2) was 399. A summary of patients' characteristics at admission to hospital is shown in Table 1.

**Table 1.** Baseline, clinical and analytical data of patients at admission to hospital due to covid-19 infection and comparison according to need of high-flow oxygen support (invasive or non-invasive ventilation, nasal high-flow cannula or non-rebreathing oxygen mask with reservoir).

| | Overall (N = 430) | High-flow oxygen support | | p value |
| --- | --- | --- | --- | --- |
| | | Yes (N = 135) | No (N = 295) | |
| **Male sex** | 249 (57.9%) | 85 (63.0%) | 164 (55.6%) | 0.091 |
| **Age, years** | 57.9±13.1 | 61.0±12.1 | 56.5±13.3 | 0.001 |
| **Comorbidies** | | | | |
| Arterial Hypertension | 169 (39.3%) | 64 (47.4%) | 105 (35.6%) | 0.013 |
| Diabetes mellitus | 70 (16.3%) | 31 (23.0%) | 39 (13.2%) | 0.009 |
| Obesity | 115 (26.9%) | 45 (33.8%) | 70 (23.8%) | 0.021 |
| Asthma | 18 (4.2%) | 8 (5.9%) | 10 (3.4%) | 0.168 |
| COPD | 23 (5.3%) | 14 (10.4%) | 9 (3.1%) | 0.003 |
| Smoker (active or former) | 104 (24.4%) | 40 (30.1%) | 64 (21.8%) | 0.045 |
| Immunosuppresive therapy | 24 (5.6%) | 13 (9.6%) | 11 (3.7%) | 0.015 |
| **Fever*** | 342 (79.5%) | 113 (83.7%) | 229 (77.6%) | 0.092 |
| **Days from fever onset** | 6.3±4.2 | 5.1±3.7 | 6.9±4.3 | <0.001 |
| **Cough** | 335 (77.9%) | 108 (80.0%) | 227 (76.9%) | 0.282 |
| **Days from cough onset** | 7.1±4.7 | 6.1±3.7 | 7.5±5.0 | 0.008 |
| **Dyspnea** | 249 (57.9%) | 92 (68.1%) | 157 (53.2%) | 0.002 |
| **Days from dyspnea onset** | 3.7±3.9 | 3.2±6.7 | 3.9±4.1 | 0.200 |
| **Diarrhea** | 135 (31.6%) | 33 (24.4%) | 102 (34.9%) | 0.019 |
| **Myalgia** | 123 (28.7%) | 31 (23.0%) | 92 (31.3%) | 0.048 |
| **Anosmia** | 36 (8.4%) | 8 (6.0%) | 28 (9.6%) | 0.146 |
| **X-ray** | | | | |
| Peripheral infiltrate | 198 (46.1%) | 36 (26.7%) | 162 (54.9%) | <0.001 |
| Unilateral pneumonia | 77 (17.9%) | 17 (12.6%) | 60 (20.3%) | |
| Bilateral pneumonia | 155 (36.0%) | 82 (60.7%) | 73 (24.7%) | |
| **Oxygen requirement at admission** | | | | |
| No | 297 (69.1%) | 63 (46.7%) | 234 (79.3%) | <0.001 |
| 0.24–0.35 L/min | 79 (18.4%) | 27 (20.0%) | 52 (17.6%) | |
| ≥0.35 L/min | 54 (12.6%) | 45 (33.3%) | 9 (3.1%) | |
| **SpO2/FiO2** | 399±108 | 323±149 | 434±55 | <0.001 |
| **Hemoglobin, mg/dL$** | 13.6±1.8 | 13.3±1.8 | 13.7±1.8 | 0.050 |
| **Total leukocytes, x10E9/L$** | 7.3±3.8 | 7.7±3.9 | 7.2±3.7 | 0.305 |
| **Lymphocyte account, x10E9/L$** | 1.1±0.7 | 1.0±0.7 | 1.2±0.6 | 0.002 |
| **Neutrophils account, x10E9/L$** | 5.5±3.2 | 6.2±3.8 | 5.3±2.9 | 0.030 |
| **Neutrophils/ Lymphocyte ratio$** | 6.5±7.0 | 8.5±10.1 | 5.6±4.8 | <0.001 |
| **Platelets, x10E9/L** | 218.4±88.4 | 207.1±82.6 | 223.6±90.6 | 0.073 |
| **INR&** | 1.2±0.4 | 1.16±0.31 | 1.18±0.43 | 0.662 |
| **D-Dimer, ng/mL#** | 748.7±3045.2 | 702.2±1431.0 | 769.2±3530.8 | 0.844 |
| **Ferritin, ng/mL€** | 831.0±776.8 | 968.4±906.0 | 776.2±713.5 | 0.037 |
| **AST, IU/mL** | 43.5±41.8 | 55.4±37.6 | 50.4±41.2 | 0.237 |
| **C reactive protein, mg/dL#** | 11.9±9.5 | 16.1±11.6 | 10.1±7.8 | <0.001 |
| **LDH, IU/mLγ** | 363.6±157.1 | 422.8±229.3 | 340.1±108.9 | <0.001 |
| **Urea, mg/dL#** | 38.7±27.9 | 48.0±35.9 | 34.5±22.2 | <0.001 |

(*Continued*)

**Table 1.** (Continued)

| | Overall (N = 430) | High-flow oxygen support | | p value |
|---|---|---|---|---|
| | | Yes (N = 135) | No (N = 295) | |
| IL-6, pg/mL & | 114.6±679.6 | 208.4±1119.9 | 72.0±315.7 | <0.001 |

* Define as temperature >38˚

$ This data was missing in 3 patients

& This data was missed in 36 patients

# This data was missing in 50 patients

γ This data was missing in 106 patients

€ This data was missing in 83 patients.

COPD, Chronic obstructive pulmonary disease.

## High-flow-oxygen-support

As aforementioned, in order to select patients who may benefit from a more intensive initial treatment, two groups were considered according to requirement or not of high-flow oxygen support at any time during hospitalization. Baseline features at admission to hospital are summarized in Table 1. Overall, up to 135 patients (31.4%) needed *high-flow-oxygen-support*, 28 (20.9%) within the first 24 hours of hospitalization. Median time from hospital admission to requirement of high-flow oxygen support was 3 days (IQR 1–5), with the majority of patients requiring 2 (N = 61) or 3 (N = 37) high-flow oxygen support devices sequentially (S1 Fig).

Patients with high-flow oxygen requirement presented with a shorter period of symptoms like fever or cough (p<0.001 and 0.008, respectively). Yet, time from the beginning of dyspnoea was similar between the two cohorts (p = 0.200). Non-respiratory symptoms, like diarrhoea and myalgia were less frequently reported in those who later required high-flow oxygen support (p = 0.019 and 0.048, respectively).

Low SpO2/FiO2 ratio at admission was associated to early requirement of *high-flow-oxygen-support*. Up to 79% of patients who did not require oxygen support at the emergency department would not need *high-flow-oxygen-support* during hospitalization. Importantly, radiologic findings were also predictive of poor prognosis, with up to 60% of patients with high-flow oxygen support presenting to hospital with bilateral pneumonia. Although patients with bilateral pneumonia were more prone to present at hospital with lower SpO2/FiO2, up to 53.5% of patients with baseline bilateral pneumonia did not need oxygen support at admission to hospital due to SpO2 >95%. S3 Table summarized the need of oxygen supplementation at admission to the ER department and the mean SpO2/FiO2 according to the baseline chest X-ray findings.

Regarding laboratory findings, patients who needed high-flow oxygen support had a lower lymphocyte, higher neutrophil count as well as a higher neutrophil/lymphocyte ratio (p = 0.002, 0.030 and <0.001, respectively). Individuals with *high-flow-oxygen-support* presented with higher levels of ferritin (968.4 vs 776.2ng/mL, p = 0.037), CRP (16.1 vs 10.1mg/dL, p<0.001), LDH (422.8 vs 340.1, p<0.001), urea (48.0 vs 34.5mg/dL, p<0.001), and IL-6 (208.4 vs 72.0pg/mL, p<0.001).

## Treatment, hospitalization and mortality

Treatment characteristics are reflected in S1 Table. Throughout the pandemic outbreak, standard of care was LPV/r, HCQ alone or in combination. Thirty-eight (8.8%) patients did not receive any of these treatments for different reasons (i.e. allergy label, intolerance or

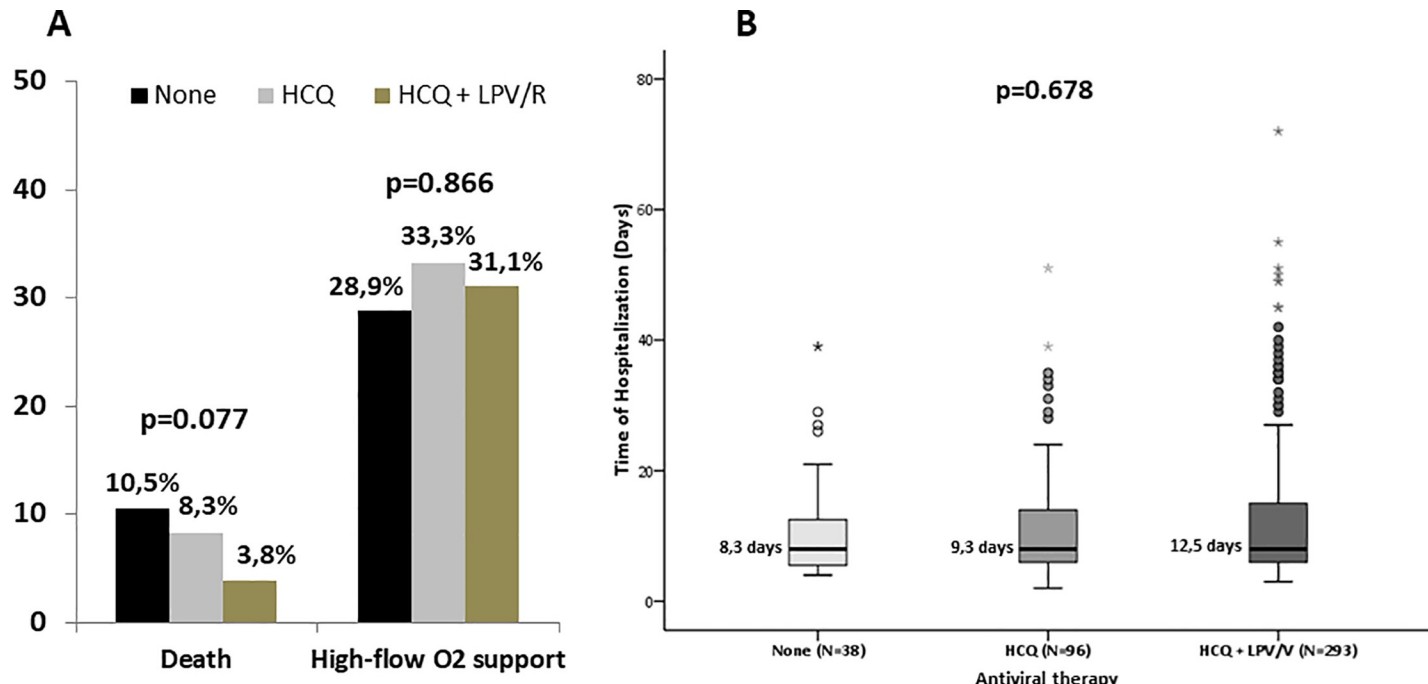

**Fig 1.** Mortality rate (panel A) and time of hospitalization (panel B) according to the antiviral therapy received.

pharmacologic interactions). The need of high-flow oxygen support requirement was not impacted by the treatment (Fig 1A).

All patients included had at least one available value of serum IL-6. Two hundred and seventy-seven (64.4%) presented at any point IL-6 values > 40 pg/dL, though only 120 patients (27.9%) received treatment with TCZ since criteria for this treatment in our centre was also SaFi < 250. Among the 277 patients with at least one value of IL-6 > 40 pg/dL, 123 (44.4%) and 18 (6.5%) required *high-flow oxygen support* or died during admission, because of the COVID-19 infection, respectively. There was no difference in the mortality rate according to the administration or not of TCZ (7.8% vs 5.6%, p = 0.315).

Fifty three (12.3%) subjects received therapy with corticosteroid at doses higher than 1 mgr/kg for 3 or more days. The relatively low number of patients treated with corticosteroids can be explained by the fact that this drug was recommended by the COVID-19 Committee of our centre only in selected cases when this study was performed. Among these patients, 38 (71.7%) required high-flow oxygen support. In order to assess the possible impact of corticosteroids therapy, time from admission to hospital to the highest FiO2 needed by each patient was calculated.

Moreover, a new variable named "early corticosteroids" was defined in those who began corticosteroids prior to the requirement of their highest FiO2. The vast majority of subjects were already on their highest FiO2 support when corticosteroids were initiated (39/53, 73.6%). Interestingly, the rate of patients who finally required *high-flow oxygen support* was statistically lower in those who underwent "early corticosteroids" (50% vs 79.5%, p = 0.042) when contrasted to those who received this therapy when they were already on their highest FiO2 requirement. In our opinion, this data supports the need of early administration of corticosteroids in order to modify the outcome of patients with COVID-19 infection.

Since only patients with severe COVID-19 have access to anti-inflammatory therapy, more patients in the group of high-flow oxygen support were treated with immunomodulatory

therapy like anti-IL-6R therapy (62.2% vs 12.2%; p<0.001) or corticosteroids pulses (28.1% vs 5.1%; p <0.001), but this difference is due to a selection bias of the patients that were more severely ill. Although treatment had no effect on the need of high-flow oxygen support (p = 0.866), there was a tendency to lower mortality among those treated with the combination of LPV/r plus HCQ (p = 0.077) (Fig 1A). Thus, antiviral treatment did not impact on the days of hospitalization (p = 0.678), as shown in Fig 1B.

Median days of hospitalization were 12.2, with longer hospitalization in those with high-flow oxygen support (25.3 vs 8.0; p<0.001). Overall mortality rate was 7.9% (34 deaths), being higher in patients with high-flow oxygen support (22.2% vs 1.4%; p<0.001). When only mortality directly related to COVID-19 was considered, the same difference was observed (overall 5.6%; *high-flow-oxygen-support* 15.6% vs 1%, p<0.001). Univariate and multivariate analysis associated with mortality due to SARS-CoV-2 are shown in S2 Table.

### Predictive models associated with *high-flow-oxygen-support* requirement

In order to help in the identification of patients at risk of poor prognosis defined as need of *high-flow-oxygen-support* during hospitalization, novel predictive models were constructed. As shown in the S1 Fig, despite the fact a few baseline factors reached a significant ROC for identification of *high-flow-oxygen-support*, only very few, such as IL-6, presented a moderate power of discrimination.

Considering the factors from the multivariate analysis from our cohort (training cohort, shown in Table 2), a model with a ROC of 0.800 was built based on 1000-bootstrap samples in order to minimize bias. As shown in Table 3 this first model included only simple baseline factors such as obesity, immunosuppressive therapy, chest x-ray findings, SpO2/FiO2, platelets and IL-6 at hospital admission. Adjusted coefficients for these variables in order to simplify their use turned out to have the same power of discrimination for the model (R = 0.800). The cut-off of 11 points presented the highest diagnostic accuracy (73.2%), with a sensitivity of 45.1%, specificity of 94.8%, negative and positive predictive value of 79.1% and 79.7%, respectively (Table 4). Inclusion of follow up laboratory parameters instead of baseline for those patients who have not achieved by that time the high-flow oxygen support outcome, led to a second predictive model with a ROC of 0.877 (Fig 2), where the only predictive factors were the baseline chest X-ray and the increase in both ferritin and IL-6 levels from admission to day 3–5 (Table 4). The cut-off of 12 points in this model showed an overall diagnostic accuracy of 86.4%, with a very high negative predictive value (92%) as shown in Table 5.

As shown in Table 1, at admission to hospital 297 (69.1%) of the patients presented SpO2 >95% and therefore they did not need oxygen support at that time. However, 63 (21.2%) of them required *high-flow-oxygen-support* later on during admission to hospital. In this subgroup of patients, the univariate analysis showed an association between the following factors and later need of *high-flow-oxygen-support*: diabetes, obesity, days of fever, days of dyspnea, chest X-ray, baseline lymphocyte and platelets account, fibrinogen and urea. However, the multivariate analysis revealed that baseline chest x-ray was the only predictor independently associated with later poor prognosis (HR 2.533, 95% CI 1.411–4.550; p = 0.002).

## Discussion

COVID-19, a novel coronavirus disease spread massively throughout the world in few months, has been responsible for millions of cases and thousands of deaths worldwide. Roughly 6% of infected patients will progress to severe disease, and, in our study up to 30% of these required high-flow oxygen support, compromising the availability of health resources. In this setting,

**Table 2. Multivariate COX proportional regression analysis of need of high-flow oxygen support in the training cohort.**

| | Multivariate analysis | |
|---|---|---|
| | HR (95% CI) | p value |
| **Male sex** | -- | 0.873 |
| **Age, years** | -- | 0.705 |
| **Arterial Hypertension** | -- | 0.327 |
| **Diabetes mellitus** | -- | 0.464 |
| **Obesity** | 2.393 (1.103–5.193) | 0.027 |
| **COPD** | -- | 0.192 |
| **Smoker (active or former)** | -- | 0.438 |
| **Immunosuppresive therapy** | 4.008 (1.331–12.073) | 0.014 |
| **X-ray** (peripheral infiltrate as reference) | | |
| Unilateral pneumonia | 2.057 (1.410–3.001) | <0.001 |
| Bilateral pneumonia | | |
| **SpO2/FiO2** | 3.495 (2.012–6.070) | <0.001 |
| **Fever** | -- | 0.535 |
| **Dyspnea** | -- | 0.596 |
| **Diarrhea** | -- | 0.061 |
| **Mialgya** | -- | 0.467 |
| **Hemoglobin, mg/dL** | -- | 0.430 |
| **Neutrophils/ Lymphocyte ratio** | -- | 0.483 |
| **Platelets, x10E9/L** | 0.436 (0.216–0.882) | 0.021 |
| **Ferritin, ng/mL** | -- | 0.645 |
| **C reactive protein, mg/dL** | -- | 0.485 |
| **LDH, IU/dL** | -- | 0.998 |
| **Urea, mg/dL** | -- | 0.608 |
| **IL-6, pg/mL** | 2.891 (1.430–5.843) | 0.003 |

early identification of patients at risk of developing a severe state is essential in order to apply treatments that could potentially impact on the disease progression.

To date, a few indexes are used [15], few of them specifically developed for the COVID-19 infection [16, 17], especially in terms of mortality. COVID-19 patients can develop different conditions that can lead to a critical situation, such as coagulopathy, thrombosis and renal failure, though clearly the most frequent presentation is respiratory failure. In this study, we developed two clinical risk scores to ascertain the need of high-flow oxygen support for COVID-19 patients. One based exclusively on baseline features and a second one based on baseline plus follow up laboratory findings. Both scores have a high diagnostic accuracy and, importantly, negative and positive predictive values for identification of patients at risk of high-flow oxygen support and, therefore, that would need an early anti-inflammatory or immunomodulatory treatment. Interestingly, the most clinically relevant factor of both scores is the chest x-ray at hospital admission.

One of the distinctive aspects and strengths of our study is the categorization of comorbidities. Previous models give the same weight to all comorbidities [16, 17], losing granularity and potentially leading to a bias since it has been postulated that not all underlying conditions increase the risk of poor prognosis in patients with COVID-19, with obesity, chronic obstructive pulmonary disease and immunosuppression as the most commonly associated with high morbidity and mortality [18].

**Table 3. Adjusted coefficients from the baseline predictive model for requirement of high-flow oxygen support using the 1000 bootstrap samples calibration cohort.**

| | HR (95% CI) | p value | Adjusted coefficient for the predictive model |
|---|---|---|---|
| **SpO2/FiO2 at admission** | | | |
| ≥300 | 1 | | 1 |
| 200–300 | 4.098 (2.412–6.962) | | 4 |
| 100–200 | 8.196 (4.824–13.924) | <0.001 | 8 |
| <100 | 12.294 (7.236–20.886) | | 12 |
| **Immunosuppressive therapy** | | | |
| No | 1 | | 1 |
| Yes | 4.100 (1.574–10.679) | 0.004 | 4 |
| **IL-6 at admission, ng/mL** | | | |
| <70 pg/mL | 1 | | 1 |
| ≥70 pg/mL | 2.165 (1.271–3.688) | 0.005 | 2 |
| **Chest x-ray at admission** | | | |
| Peripheral infiltrates | 1 | | 1 |
| Unilateral pneumonia | 1.958 (1.463–2.621) | <0.001 | 2 |
| Bilateral pneumonia | 3.916 (2.926–5.242) | | 4 |
| **Obesity** | | | |
| No | 1 | | 1 |
| Yes | 1.782 (1.004–3.163) | 0.048 | 2 |
| **Platelets at admission** | | | |
| ≤215x10E9/mL | 1 | | 1 |
| >215x10E9/mL | 0.551 (0.322–0.944) | 0.030 | 0.5 |

In our cohort, although some comorbidities such as obesity, diabetes, hypertension, COPD and immunosuppression were associated with a higher rate of high-flow oxygen support, only obesity and immunosuppressive therapy were independently linked with the composite end-point. Interestingly, clinical presentation differed in patients that later required or not of high-flow oxygen support. Dyspnoea was more commonly reported by those who needed this oxygen support, but non-respiratory findings such as myalgias and diarrhoea were more frequent among patients with good prognosis. In fact, the possible positive impact of diarrhoea in patients with COVID-19 should be highlighted, since a tendency towards higher frequency of this symptom was observed among patients without high-flow oxygen requirement. It may be

**Table 4. Adjusted coefficients from the baseline and evolutionary predictive model for requirement of high-flow oxygen support using the 1000 bootstrap samples calibration cohort.**

| | HR (95% CI) | p value | Adjusted coefficient for the predictive model |
|---|---|---|---|
| **Increase in Ferritin levels** | | | |
| <200 ng/mL | 1 | | 1 |
| ≥200 ng/mL | 7.486 (2.620–21.387) | | 7.5 |
| **Increase in IL-6 levels** | | | |
| <64 pg/mL | 1 | | 1 |
| ≥64 pg/mL | 18.088 (6.001–54.519) | <0.001 | 18 |
| **Chest x-ray at admission** | | | |
| Peripheral infiltrates | 1 | | 1 |
| Unilateral pneumonia | 2.130 (1.195–3.796) | 0.002 | 2 |
| Bilateral pneumonia | 4.260 (2.390–7.592) | | 4 |

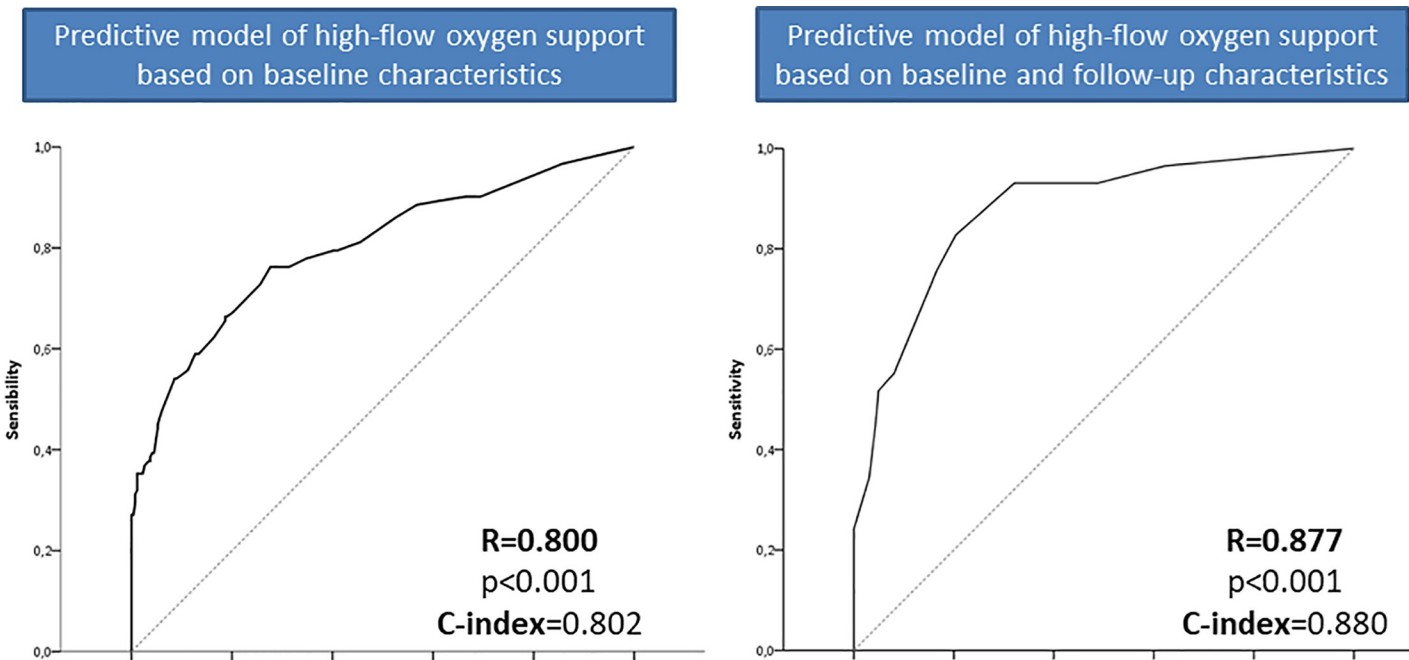

**Fig 2. Area under the receiver operating characteristic (AUROC) of the models for prediction of need of high-flow oxygen support.**

worth exploring this negative association on an independent cohort with a larger number of patients.

Despite the fact many laboratory parameters have been previously related to poor outcomes, such as the neutrophils/lymphocyte ratio, urea, LDH, platelets and acute phase reactants such as ferritin, procalcitonin and CRP [2], the analytical backbone of our models was the IL-6. The importance of this interleukin in the cytokine release syndrome mediated by the COVID-19 is well known [4].

Interestingly, in the follow up model the only factors independently associated with the need of high-low oxygen support were chest x-ray and increase in inflammatory parameters such as IL-6 and ferritin from day 0 and days 3–5, yielding to a model with a high performance (R = 0.877). This emphasizes the importance of inflammatory determinants at baseline and throughout the course of the disease. As aforementioned, the chosen outcome of our predictive models was the requirement of high-flow oxygen support.

**Table 5. Accuracy of the novel models for prediction of patients at risk of high-flow oxygen support.**

| | Baseline model | Baseline and evolutionary model |
|---|---|---|
| **Enrolled patients** | **390** | **191** |
| **ROC** | **0.800** | **0.877** |
| **Cut-off value** | **11** | **12** |
| **Sensitivity, % (95% CI)** | 45.1 (36.5–53.9) | 55.2 (37.6–72.6) |
| **Specificity, % (95% CI)** | 94.8 (91.4–96.9) | 92.0 (86.8–95.3) |
| **Negative predictive value, % (95% CI)** | 79.1 (74.4–83.2) | 92.0 (86.8–95.3) |
| **Positive predictive value, % (95% CI)** | 79.7 (68.8–87.5) | 55.2 (37.6–72.6) |
| **Diagnostic accuracy, % (95% CI)** | 73.2 (74.9–83.0) | 86.4 (80.8–90.5) |

Although previous studies have mainly focused on COVID-19 mortality and/or intensive care unit (ICU) admission [16, 19], the election of our composite endpoint was based on the real-world situation during the COVID-19 pandemic peak, when ICU beds, NHFC or even non-invasive support were overwhelmed by the huge number of patients with severe COVID-19. Though non-rebreathing oxygen mask with reservoir was the most commonly used support, the majority of patients required sequentially two or more devices, a fact that emphasized the severity of patients included in the cohort.

Our study presents some limitations. This is a retrospective and unicentric cohort. Though characterization of comorbidities was thorough, predictive factors in the predictive models varied due to the different number of patients in both models, due to loss of patients (those who met the outcome prior to the analytical follow-up) and also, but to a lesser extent, due to lack of follow up IL-6 or ferritin values in some patients.

Nevertheless, the models were calibrated by 1000-bootstrap samples, and the main factors associated with the need of high-flow oxygen support were consistent in both models (baseline chest x-ray and IL-6). Another strength of our study is the still low number of reports on COVID-19 predictive prognostic scores in non-Asian population.

Moreover, in our study the outcome of patients has been assessed in terms of respiratory support, including not only invasive mechanical ventilation, but all types of high-flow devices, leading to a more realistic overview of the COVID-19 severity than those scores just focusing on the need for ICU admission or mortality. Since the SARS-CoV-2 had just been identified at the time of our study, kinetics of viral load and antibody titres anti-SARS-CoV-2 were not available.

## Conclusions

In conclusion, a third of patients admitted to the hospital with COVID-19 needed high-flow oxygen support which is associated with a worse prognosis. This study provides predictive models for disease severity based on baseline characteristics, allowing an early identification of those patients who can benefit from a more intensive disease management in order to prevent or reduce COVID-19 complications.

## Supporting information

**S1 Fig. Area under the receiver operating characteristic (AUROC) for prediction of need of high-flow oxygen support of the main characteristics, symptoms and laboratory parameters from patients admitted to hospital due to COVID-19 pneumonia: Age, days from fever onset to admission to hospital, SpFi and ferritin.**
(TIF)

**S2 Fig. Area under the receiver operating characteristic (AUROC) for prediction of need of high-flow oxygen support of the main laboratory parameters from patients admitted to hospital due to COVID-19 pneumonia: Neutrophils/Lymphocytes (N/L) ratio, LDH, IL-6 and D-dimer.**
(TIF)

**S1 Table. Treatment and clinical endpoints according to the high-flow oxygen support (invasive or non-invasive ventilation, nasal high-flow cannula or non-rebreathing oxygen mask with reservoir).**
(PDF)

**S2 Table. Univariate and multivariate analysis of factors associated with mortality rate related to SARS CoV-2.**
(PDF)

**S3 Table. Number of patients with need of oxygen supplementation at admission to the ER department and the mean SpO2/FiO2 according to the baseline chest X-ray findings.**
(PDF)

## Acknowledgments

We thank the entire clinical staff who took care of the patients during the COVID-19 outbreak situation and all the members of the "Vall d'Hebron COVID-19 immunological profile study group". English writing support was provided by Fidelma Greaves.

## Author Contributions

**Conceptualization:** Mar Riveiro-Barciela, Moisés Labrador-Horrillo, Fernando Martínez-Valle.

**Formal analysis:** Moisés Labrador-Horrillo.

**Investigation:** Mar Riveiro-Barciela, Laura Camps-Relats, Didac González-Sans, Meritxell Ventura-Cots, María Terrones-Peinador, Andrea Nuñez-Conde, Ricardo Pujol-Borrell.

**Methodology:** Mónica Martínez-Gallo, Manuel Hernández, Andrés Antón, Fernando Martínez-Valle.

**Resources:** Manuel Hernández, Andrés Antón, Ricardo Pujol-Borrell.

**Supervision:** Mar Riveiro-Barciela, Fernando Martínez-Valle.

**Writing – original draft:** Mar Riveiro-Barciela, Moisés Labrador-Horrillo.

**Writing – review & editing:** Mar Riveiro-Barciela, Moisés Labrador-Horrillo, Laura Camps-Relats, Didac González-Sans, Meritxell Ventura-Cots, María Terrones-Peinador, Andrea Nuñez-Conde, Mónica Martínez-Gallo, Manuel Hernández, Andrés Antón, Antonio González, Ricardo Pujol-Borrell, Fernando Martínez-Valle.

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
