## [Decision Letter · Decision Letter 0]

8 Nov 2020

PONE-D-20-30311

Simple predictive models identify patients with COVID-19 pneumonia and poor prognosis

PLOS ONE

Dear Dr. Labrador-Horrillo,

Thank you for submitting your manuscript to PLOS ONE. After careful consideration, we feel that it has merit but does not fully meet PLOS ONE’s publication criteria as it currently stands. Therefore, we invite you to submit a revised version of the manuscript that addresses the points raised during the review process.

We look forward to receiving your revised manuscript.

Kind regards,

Tai-Heng Chen, M.D.

Academic Editor

PLOS ONE

1- Please ensure that your manuscript meets PLOS ONE's style requirements, including those for file naming. The PLOS ONE style templates can be found at

2. In the ethics statement in the manuscript and in the online submission form, please provide additional information about the patient records used in your retrospective study, including: a) whether all data were fully anonymized before you accessed them; b) the date range (month and year) during which patients' medical records were accessed. If patients provided consent to have data from their medical records used in research, please include this information, including: whether consent was informed and what type you obtained (for instance, written or verbal, and if verbal, how it was documented and witnessed).

3. We note you have included a table to which you do not refer in the text of your manuscript. Please ensure that you refer to Table 5 in your text; if accepted, production will need this reference to link the reader to the Table.

Reviewers' comments:

Reviewer's Responses to Questions

**Comments to the Author**

1. Is the manuscript technically sound, and do the data support the conclusions?

Reviewer #1: No

Reviewer #2: Yes

Reviewer #3: Yes

2. Has the statistical analysis been performed appropriately and rigorously? 

Reviewer #1: Yes

Reviewer #2: Yes

Reviewer #3: Yes

3. Have the authors made all data underlying the findings in their manuscript fully available?

Reviewer #1: Yes

Reviewer #2: Yes

Reviewer #3: Yes

4. Is the manuscript presented in an intelligible fashion and written in standard English?

Reviewer #1: Yes

Reviewer #2: Yes

Reviewer #3: Yes

5. Review Comments to the Author

Reviewer #1: This study aimed to establish predictive models for COVID-19 pneumonia severity in hospitalized patients.

The authors attempted to build predictive models to identify the patients who required high-flow-oxygen-support .

My major concerns are that low SpO2/FiO2 and abnormal chest x-ray(high percentage of bilateral pneumonia) are significant factors in the models.

In the causal model, significant pneumonia may lead to low SpO2/FiO2, which may increase the use of high flow oxygen support. SpO2/FiO2 would be an intermediate variable which should not be added in the models for adjustment. I suggest the authors build the predictive models without SpO2/FiO2 to evaluate if the AUC is still acceptatble.

Reviewer #2: The article ‘Simple predictive models identify patients with COVID-19 pneumonia and poor prognosis' presents a predictive model for COVID-19 pneumonia in hospitalized patients. The predictive model showed a ROC of 0.800 based on: SpO2/FiO2 (adjusted Hazard Ratio-aHR:8), chest x-ray (aHR:4), prior immunosuppressive therapy (aHR:4), obesity (aHR:2), IL-6 (aHR:2), platelets (aHR:0.5) by the analysis of 430 patients admitted in Vall d’Hebron Hospital (Barcelona). This is a very important data in the pandemic of COVID-19. My comments are below:

Major Comments

1. Criteria of induction of high-flow oxygen support

Results included some analysis of the patient's background using high-flow oxygen support. For example, they were bilateral abnormal shadows in a chest X-ray image and a decrease in the number of lymphocytes. On the other hand, since the results are from a single center, there may be a protocol for induction of high-flow oxygen support. First, please describe the criteria for introduction of high-flow oxygen support. The information will be important information for evaluating the predictive models.

２．Predictive score of induction of high-flow oxygen support in COVID-19

　Age, medical past history (hypertension, diabetes, obesity, COPD, smoking, immunosuppressive therapy), NLR, CRP, ferritin and LDH were with significant difference in induction of high-flow oxygen support. Is it possible to statistically calculate the predictive score of whether to introduce high-flow oxygen support in COVID-19? We would like to ask if it is possible to predict X% for 3 out of 6, Y% for 4, Z% for 5, and so on.

Reviewer #3: Authors conducted a retrospective study of 430 patients admitted in Vall d’Hebron Hospital (Barcelona) between 03-12-2020 and 04-28-2020 due to COVID-19 pneumonia and would like to create a model to identify risk factors in patients with high-flow-oxygen-support group, they identify SpO2/FiO2 and chest x-ray on admission or changes on inflammatory parameters as IL-6 and ferritin allow us early identification of COVID-19 patients at risk of high-flow-oxygen-support that may benefit from a more intensive disease management.

Some concerns need to clarify.

Major

1. Authors said 135 patients (31.4%) needed high-flow-oxygen-support after admission. How many patients who initially had SpO2 >95% but finally needed High-flow-oxygen-support, any risk factors about progress to desaturation in the subgroup? Associated with treatment options?

2. How to distinguish “peripheral infiltrate” from “unilateral pneumonia “or “bilateral pneumonia”? in clinical, peripheral infiltrate belongs to the above. Do authors define pneumonia is consolidation or air space shadow instead of infiltrates?

3. Figure 1A. Why up to 28.9% patients with high flow oxygen support didn’t receive any therapy such as LPV/r plus HCQ or HCQ in your cohort, the treatment options may seriously affect outcome and made the risk factors of poor prognosis became not reliable. Authors should compare the efficacy in different treatment subgroups in patients with high-flow-oxygen-support.

4. How to decide these patients with high-flow-oxygen-support had to receive LPV/r plus HCQ or HCQ or none? In figure 1B, authors showed the time of hospitalization in different treatment arms, however, the time may be influenced by death. Therefore, authors are encouraged to define other endpoints, such as the time to leave High-flow-oxygen-support or successful discharge.

5. Please define the “obesity” since obesity is a significant risk factor in multivariable COX regression model in the study.

6. The role of serum IL-6 in your cohort? A prognostic factor or indicator for drug selection? Any different outcome in patients with IL-6 higher than 40pg/dL and SpO2/FiO2 (SaFi) below 250 received or not received TCZ in your cohort? All high serum IL-6 patients received TCZ, or combination therapy?

7. How many patients received corticosteroid in your cohort since several studies in JAMA and NEJM indicated corticosteroid administration may have better outcome.

8. No patients received redemsivir, the only FDA approval emergent use drug for COVID-19, in your cohort?

Minor

Page 6, line 21, hidroxychloroquine  hydroxychloroquine

Page 8, line 14, chronic pulmonary disease  chronic obstructive pulmonary disease

6. PLOS authors have the option to publish the peer review history of their article (what does this mean?). If published, this will include your full peer review and any attached files.

Reviewer #1: No

Reviewer #2: **Yes: **Naoyuki Matsuda M.D., Ph.D

Reviewer #3: No

---

## [Author Response · Author response to Decision Letter 0]

17 Nov 2020

#Journal requirements:

1- Please ensure that your manuscript meets PLOS ONE's style requirements, including those for file naming. The PLOS ONE style templates can be found at

We have ensured that our manuscript meets PLOS ONE's style requirements

2. In the ethics statement in the manuscript and in the online submission form, please provide additional information about the patient records used in your retrospective study, including: a) whether all data were fully anonymized before you accessed them; b) the date range (month and year) during which patients' medical records were accessed. If patients provided consent to have data from their medical records used in research, please include this information, including: whether consent was informed and what type you obtained (for instance, written or verbal, and if verbal, how it was documented and witnessed).

We have corrected the ethics statement accordingly and added it in the manuscript.

3. We note you have included a table to which you do not refer in the text of your manuscript. Please ensure that you refer to Table 5 in your text; if accepted, production will need this reference to link the reader to the Table.

Table 5 is correctly cited in the “Results” section located now in Page 13 line 17-18. We have highlighted it in yellow in the manuscript.

We added “Supporting Information” files at the end of the manuscript.

#Reviewers' comments:

Reviewer #1:

This study aimed to establish predictive models for COVID-19 pneumonia severity in hospitalized patients. The authors attempted to build predictive models to identify the patients who required high-flow-oxygen-support. My major concerns are that low SpO2/FiO2 and abnormal chest x-ray (high percentage of bilateral pneumonia) are significant factors in the models. In the causal model, significant pneumonia may lead to low SpO2/FiO2, which may increase the use of high flow oxygen support. SpO2/FiO2 would be an intermediate variable which should not be added in the models for adjustment. I suggest the authors build the predictive models without SpO2/FiO2 to evaluate if the AUC is still acceptable.

We appreciate the reviewer’s suggestion. Although patients with bilateral pneumonia were more prone to present at hospital with lower SpO2/FiO2, up to 53.5% of patients with baseline bilateral pneumonia did not need oxygen support at admission to hospital due to SpO2 >95%. A new S3 Table summarized the need of oxygen supplementation at admission to the ER department and the mean SpO2/FiO2 according to the baseline chest X-ray findings.

This data has been added to the revised version of the manuscript in order to improve the quality of the paper (Page 10 lines 11 – 15).

Regarding the causal model, though as previously mentioned chest X-ray was associated with lower SpO2/FiO2, both were independent factors associated with the need of high-flow oxygen support, and for this reason they were included in the predictive model. However, as suggested by the reviewer, we conducted a new multivariate analysis excluding the variable “SpO2/FiO2”, turning out the following predictive model:

This model yields a ROC of 0.710 (0.650-0.770), value lower than the ROC showed by the predictive model including SpO2/FiO2 (ROC 0.800).

This new analysis may be added to the manuscript as supporting information data if the reviewer or editor thinks it would be suitable for readers.

 

Reviewer #2:

The article ‘Simple predictive models identify patients with COVID-19 pneumonia and poor prognosis' presents a predictive model for COVID-19 pneumonia in hospitalized patients. The predictive model showed a ROC of 0.800 based on: SpO2/FiO2 (adjusted Hazard Ratio-aHR:8), chest x-ray (aHR:4), prior immunosuppressive therapy (aHR:4), obesity (aHR:2), IL-6 (aHR:2), platelets (aHR:0.5) by the analysis of 430 patients admitted in Vall d’Hebron Hospital (Barcelona). This is a very important data in the pandemic of COVID-19. My comments are below:

1. Criteria of induction of high-flow oxygen support

Results included some analysis of the patient's background using high-flow oxygen support. For example, they were bilateral abnormal shadows in a chest X-ray image and a decrease in the number of lymphocytes. On the other hand, since the results are from a single center, there may be a protocol for induction of high-flow oxygen support. First, please describe the criteria for introduction of high-flow oxygen support. The information will be important information for evaluating the predictive models.

Thank you very much for your comment. Related to the oxygen support, the target was to obtain an oxygen saturation higher than 90-92%, or a SaFi higher than 250. In this sense, when the patient required a FiO2 higher than 50%, a non-rebreathing oxygen mask with reservoir, nasal high-flow cannula, non-invasive mechanical ventilation, and invasive mechanical ventilation were used sequentially. To clarify this point a sentence has been added to the manuscript: Page 6 lines 14 – 15: “These mechanisms were used in a sequential way to obtain oxygen saturation over 90% in the case that the patient required a FiO2 higher than 50%”.

2. Predictive score of induction of high-flow oxygen support in COVID-19, age, medical past history (hypertension, diabetes, obesity, COPD, smoking, immunosuppressive therapy), NLR, CRP, ferritin and LDH were with significant difference in induction of high-flow oxygen support. Is it possible to statistically calculate the predictive score of whether to introduce high-flow oxygen support in COVID-19? We would like to ask if it is possible to predict X% for 3 out of 6, Y% for 4, Z% for 5, and so on. 

We appreciate the reviewer’s comment. Since all the factors included in the multivariate analysis are quite common and frequently available, we constructed the baseline predictive model with adjusted coefficients after a 1000-bootstrap samples calibration, the latter as a strategy to avoid the bias of the lack of a validation cohort.

Absence of data of any of the factors, i.e. analytical parameters such as IL-6 and platelets, would indeed decrease the performance of the model, though it still achieves a moderate ROC, with ROC of 0.789 (95% CI 0.740-0.839) and 0.774 (95% CI 0.723-0.825), for the model without IL-6 and without both IL-6 and platelets values, respectively.

Nevertheless, since we realized that exclusion of one or more factors may change the predictive model, we calculated the new coefficients after discarding the IL-6 plus/minus platelets values (the factors that may be more usually not available), resulting in the following coefficients:

- Baseline model without IL-6: Immunosuppression * 4.372 (Hazard Ratio-HR) + Chest x-ray (3 categories) * 1.924 + SpO2/FiO2 (4 categories) * 4.593 + Obesity * 1.801 + Platelets (>215x10E9/mL) * 0.558.

- Baseline model neither IL-6 nor platelets: Immunosuppression * 3.816 + Chest x-ray (3 categories) * 1.929 + SpO2/FiO2 (4 categories) * 4.371 + Obesity * 1.714.

The ROC of the aforementioned new baseline models are shown in figures included in the letter in response to reviewers.

These new analyses may be added to the manuscript as supporting information data if the reviewer or editor thinks it would be suitable for readers.

 

Reviewer #3:

Authors conducted a retrospective study of 430 patients admitted in Vall d’Hebron Hospital (Barcelona) between 03-12-2020 and 04-28-2020 due to COVID-19 pneumonia and would like to create a model to identify risk factors in patients with high-flow-oxygen-support group, they identify SpO2/FiO2 and chest x-ray on admission or changes on inflammatory parameters as IL-6 and ferritin allow us early identification of COVID-19 patients at risk of high-flow-oxygen-support that may benefit from a more intensive disease management. Some concerns need to clarify.

1. Authors said 135 patients (31.4%) needed high-flow-oxygen-support after admission. How many patients who initially had SpO2 >95% but finally needed High-flow-oxygen-support, any risk factors about progress to desaturation in the subgroup? Associated with treatment options? 

We agree with the reviewer that analysis of patients with baseline SpO2 >95% is very interesting. As shown in Table 1, at admission to hospital 297 (69.1%) of the patients presented SpO2 >95% and therefore they didn’t need oxygen support at that time. However, 63 (21.2%) required high-flow-oxygen-support later on during admission to hospital. In line with overall results from our cohort, and more than probably due to a bias selection and late administration, patients treated with either TCZ or corticosteroids presented a statistically higher need of high-flow-oxygen-support (TCZ: 62% vs 8.4%, p<0.001; Corticosteroids: 77.4% vs 14.7%, p<0.001). Nevertheless, lack of therapy in comparison with HCQ alone or in combination with LPV/r had no impact on prognosis in terms of high-flow-oxygen-support (16.0% vs 22.2% vs 21.4%, respectively; p=0.800). 

In this subgroup of patients, the univariate analysis showed an association between the following factors and later need of high-flow-oxygen-support: diabetes, obesity, days of fever, days of dyspnea, chest CX-ray, baseline lymphocyte and platelets account, fibrinogen and urea. However, the multivariate analysis revealed that baseline chest x-ray was the only predictor independently associated with later poor prognosis (HR 2.533, 95% CI 1.411-4.550; p=0.002). 

This data has been added to the revised version of the manuscript in order to improve the quality of the paper. “Page 13 line 19 – Page 14 line 2”

2. How to distinguish “peripheral infiltrate” from “unilateral pneumonia “or “bilateral pneumonia”? in clinical, peripheral infiltrate belongs to the above. Do authors define pneumonia is consolidation or air space shadow instead of infiltrates?

We highly appreciate the comments of the reviewer. As we state in the manuscript, chest radiography was interpreted by two attending physicians. Unilateral pneumonia was defined if an infiltrate was observed in only one of the lungs. Peripheral infiltrate was described when involvement was mainly present in the subpleural area. Bilateral pneumonia was considered if diffuse infiltrates were present in both hemithorax. This explanation has been added to the main text in order to clarify concepts. Page 5 line 25 – Page 6 line 4

3. Figure 1A. Why up to 28.9% patients with high flow oxygen support didn’t receive any therapy such as LPV/r plus HCQ or HCQ in your cohort, the treatment options may seriously affect outcome and made the risk factors of poor prognosis became not reliable. Authors should compare the efficacy in different treatment subgroups in patients with high-flow-oxygen-support.

We appreciate the kind comments of the reviewer. At the time that patients were collected, between the beginning of March and the end of April 2020, the standard of care in our hospital for patients affected by COVID19 was HCQ plus LPV/r, both alone or in combination. However, pharmacological interactions were very frequent with these drugs and other agents. On the other hand, side effects, like diarrhea and QT enlargement, mostly related to LPV/r were present in many patients, and for this reason these drugs were stopped early on. Fortunately, different studies have discarded any beneficial role of these treatments in patients with COVID19, and for this reason they are no longer used.

4. How to decide these patients with high-flow-oxygen-support had to receive LPV/r plus HCQ or HCQ or none? In figure 1B, authors showed the time of hospitalization in different treatment arms, however, the time may be influenced by death. Therefore, authors are encouraged to define other endpoints, such as the time to leave High-flow-oxygen-support or successful discharge. 

We appreciated the reviewer’s observation. As summarized in the Materials and methods section and commented previously in this response, according to our centre protocol for treatment of COVID19, all admitted patients with pneumonia received treatment with LPV/r plus/minus HCQ, except contraindication, drug-drug interaction or intolerance. 

We agree with the reviewer that it would be interesting to assess the possible impact of pharmacological treatment on time to successful discharge. There was no difference in the mean time of admission according to the received treatment (None 8.3 days; HCQ 9.3 days; HCQ plus LPV/r 12.5; p=0.678)

These last data have been added to figure 1B in order to highlight successful discharge.

5. Please define the “obesity” since obesity is a significant risk factor in multivariable COX regression model in the study.

According to WHO, obesity was defined as a BMI higher than 30. This has been added to in the Materials and methods section. “Page 5 lines 19 – 20”

6. The role of serum IL-6 in your cohort? A prognostic factor or indicator for drug selection? Any different outcome in patients with IL-6 higher than 40pg/dL and SpO2/FiO2 (SaFi) below 250 received or not received TCZ in your cohort? All high serum IL-6 patients received TCZ, or combination therapy?

All patients included had at least one available value of serum IL-6. Two hundred and seventy-seven (64.4%) presented at any point IL-6 values > 40 pg/dL, though only 120 (27.9%) received treatment with TCZ since criteria for this treatment in our centre was also SaFi < 250. 

Among the 277 patients with at least one value of IL-6 > 40 pg/dL, 123 (44.4%) and 18 (6.5%) required high-flow oxygen support or died during admission, because of the COVID-19 infection, respectively. There was no difference in the mortality rate according to the administration or not of TCZ (7.8% vs 5.6%, p=0.315). However, need of high-flow oxygen support was statistically higher among those treated with TCZ in contrast with those who did not undergo this therapy (62.2% vs 12.2%, p<0.001). As discussed in the manuscript: Page 12 lines 3 – 7 “Since only patients with severe COVID-19 have access to anti-inflammatory therapy, more patients in the group of high-flow oxygen support were treated with immunomodulatory therapy like anti-IL-6R therapy (62.2% vs 12.2%; p<0.001) or corticosteroids pulses (28.1% vs 5.1%; p <0.001), but this difference is due to a selection bias of the patients that were more severely ill.” 

Once again, this fact emphasized the need of early identification and anti-inflammatory treatment of patients at risk of poor prognosis.

We added a paragraph to the manuscript to clarify the question: Page 11 lines 4 – 11.

7. How many patients received corticosteroid in your cohort since several studies in JAMA and NEJM indicated corticosteroid administration may have better outcome.

We appreciate the reviewer’s comment. Fifty three (12.3%) subjects received therapy with corticosteroid at doses higher than 1 mgr/kg for 3 or more days. The relatively low number of patients treated with corticosteroids can be explained by the fact that this drug was recommended only in selected cases by the COVID-19 Committee of our centre when this study was performed.

Among these patients, 38 (71.7%) required high-flow oxygen support. In order to assess the possible impact of corticosteroids therapy, time from admission to hospital to the highest FiO2 needed by each patient was calculated. Moreover, a new variable named “early corticosteroids” was defined in those who began corticosteroids prior to the requirement of their highest FiO2. The vast majority of subjects were already on their highest FiO2 support when corticosteroids were initiated (39/53, 73.6%). Interestingly, the rate of patients who finally required high-flow oxygen support was statistically lower in those who underwent “early corticosteroids” (50% vs 79.5%, p=0.042) when contrasted to those who received this therapy when they were already on their highest FiO2 requirement. In our opinion, this data supports the need of early onset of corticosteroid in order to modify outcome of patients with COVID-19 infection. 

We added these new data to the manuscript Page 11 line 12 – Page 12 line 2

8. No patients received redemsivir, the only FDA approval emergent use drug for COVID-19, in your cohort?

In our cohort just one patient received Remdesivir because its use was restricted in our hospital at that time, to ICU patients. This patient was admitted to ICU the same day he consulted to the ER department due to severe pneumonia that required HFNC during the first hours of admission.

Page 6, line 21, hidroxychloroquine  hydroxychloroquine

Page 8, line 14, chronic pulmonary disease  chronic obstructive pulmonary disease

We appreciate the reviewer’s comment. We have corrected these spelling errors

---

## [Decision Letter · Decision Letter 1]

15 Dec 2020

Simple predictive models identify patients with COVID-19 pneumonia and poor prognosis

PONE-D-20-30311R1

Dear Dr. Labrador-Horrillo,

We’re pleased to inform you that your manuscript has been judged scientifically suitable for publication and will be formally accepted for publication once it meets all outstanding technical requirements.

Kind regards,

Tai-Heng Chen, M.D.

Academic Editor

PLOS ONE

Reviewers' comments:

Reviewer's Responses to Questions

**Comments to the Author**

1. If the authors have adequately addressed your comments raised in a previous round of review and you feel that this manuscript is now acceptable for publication, you may indicate that here to bypass the “Comments to the Author” section, enter your conflict of interest statement in the “Confidential to Editor” section, and submit your "Accept" recommendation.

Reviewer #1: All comments have been addressed

Reviewer #3: All comments have been addressed

2. Is the manuscript technically sound, and do the data support the conclusions?

Reviewer #1: Yes

Reviewer #3: Yes

3. Has the statistical analysis been performed appropriately and rigorously? 

Reviewer #1: Yes

Reviewer #3: Yes

4. Have the authors made all data underlying the findings in their manuscript fully available?

Reviewer #1: Yes

Reviewer #3: Yes

5. Is the manuscript presented in an intelligible fashion and written in standard English?

Reviewer #1: Yes

Reviewer #3: Yes

6. Review Comments to the Author

Reviewer #1: (No Response)

Reviewer #3: All questions I raised were appropriately answered, and it can be accepted in PLOS one in current form.

7. PLOS authors have the option to publish the peer review history of their article (what does this mean?). If published, this will include your full peer review and any attached files.

Reviewer #1: No

Reviewer #3: No

---

## [Editor Report · Acceptance letter]

17 Dec 2020

PONE-D-20-30311R1 

Simple predictive models identify patients with COVID-19 pneumonia and poor prognosis. 

Dear Dr. Labrador-Horrillo:

I'm pleased to inform you that your manuscript has been deemed suitable for publication in PLOS ONE. Congratulations! Your manuscript is now with our production department. 

Kind regards, 

on behalf of

Dr. Tai-Heng Chen 

Academic Editor

PLOS ONE